# Development and Application of Potentially Universal Microsatellite Markers for Pheasant Species

**DOI:** 10.3390/ani13233601

**Published:** 2023-11-21

**Authors:** Daxin Xie, Nan Yang, Wencai Xu, Xue Jiang, Lijun Luo, Yusen Hou, Guangqing Zhao, Fujun Shen, Xiuyue Zhang

**Affiliations:** 1Key Laboratory of Bio-Resource and Eco-Environment of Ministry of Education, College of Life Sciences, Sichuan University, Chengdu 610065, China; 17361043520@163.com (D.X.);; 2Institute of Qinghai-Tibetan Plateau, Southwest Minzu University, Chengdu 610225, China; yangnan0204@126.com; 3School of Pharmacy, Chengdu University of Traditional Chinese Medicine, Chengdu 610056, China; 4Sichuan Key Laboratory of Endangered Wildlife Conservation Biology, Research Base of Giant Panda Breeding, Chengdu 610081, China

**Keywords:** Phasianidae, genome, universal microsatellite markers, analysis of population genetics

## Abstract

**Simple Summary:**

In this study, 471 microsatellite loci which are common among 8 pheasant species were screened based on genome data, and 119 loci were selected to develop microsatellite markers. After PCR amplifications and reaction condition optimizations, and validation of microsatellite loci in 14 species of 11 genera within Phasianidae. Finally, 49 potentially universal microsatellite markers in the pheasant species were obtained. The 49 potentially universal microsatellite loci were successfully applied to a genetic diversity assessment and comparison for three pheasant birds, the Sichuan Hill partridge, blood pheasant and buff-throated partridge. These 49 microsatellite loci are potentially universal microsatellite loci for pheasants and are of great significance to establish a shared platform in population genetics study of pheasants.

**Abstract:**

Pheasants are widely distributed in the southwest of China, but many of them are endangered due to habitat fragmentation and environmental changes. Genetic diversity is crucial for species to maintain their evolutionary potential, and thus it is important to develop universal genetic markers for facilitating the assessment of genetic diversity and planning effective conservation actions in these endangered species. In this study, 471 microsatellite loci which are common among eight pheasant species were screened based on genome data, and 119 loci were selected to develop microsatellite markers. After PCR amplifications and reaction condition optimizations, and validation of microsatellite loci in 14 species of 11 genera within Phasianidae. Finally, 49 potentially universal microsatellite markers in pheasant species were obtained. These microsatellite markers were successfully applied to assess the genetic diversity of 3 pheasant species. The Sichuan hill partridge (*Arborophila rufipectus*), blood pheasant (*Ithaginis cruentus*), buff-throated partridge (*Tetraophasis szechenyii*) and Sichuan hill partridge had a relatively low genetic diversity level. These 49 microsatellite loci are potentially universal microsatellite loci for pheasants and are of great significance to establish a shared platform in population genetics study of pheasants.

## 1. Introduction

Galliformes are one of the more primitive bird groups and one of the earlier differentiated groups of extant birds. Crowe used 102 morphological and behavioral parameters to construct a phylogenetic structure of the order Galliformes and proposed a five-family classification system that merges the families Turkeyidae and Grouseidae into Phasianidae, which has been widely recognized by the scientific community [1]. Pheasants have experienced rapid adaptive radiation and convergent morphological evolution, resulting in discrepancies between traditional phylogenetic relationships defined based on morphological and behavioral traits and evolutionary histories constructed based on molecular markers, and studies on the phylogeny of pheasants are still a research hotspot in the Galliforme species [1,2]. As ground-dwelling birds with poor flight and dispersal abilities, these species are sensitive to habitat disturbance and hunting pressure. Phasianidae is also the largest family of Galliformes, with a total of 54 genera and 189 species [3]. As ground-dwelling birds with poor flight and dispersal abilities, these species are sensitive to habitat disturbance and hunting pressure and have been suggested as good indicators of the overall habitat conservation status for planning effective forest management. Internationally, while the status of individual species has been assessed against the IUCN Red List Criteria, there is no overall analysis of the conservation challenges facing the pheasant family [4,5]. China has the richest distribution of wild pheasants in the world, containing 63 species in 26 genera, and most of them are distributed in southwest China. Among them, 21 pheasant species, including the Chinese Monal (*Lophophorus lhuysii*) and the Hainan partridge (*Arborophila ardens*), are endemic to China [6,7]. Most of the pheasant species are listed in the Chinese red list and 23 pheasant species are national key protected species and 28 pheasant species are national second key protected species, such as the Reeves’s pheasant (*Syrmaticus reevesii*) [8]. Due to illegal hunting, coupled with habitat decline and loss, many pheasant species are on the verge of extinction, which shows the seriousness of the conservation situation of pheasant species and the need to promote conservation research [9,10,11].

Genetic diversity is the basis for species to maintain their evolutionary potential, so the conservation of genetic diversity is crucial in species conservation [12]. Microsatellite markers have been widely used in the scientific areas of molecular ecology and population genetics of endangered animals due to their wide distribution in genome, rich polymorphic information content, and co-dominant inheritance [13,14,15,16]. For conservation ecology studies of wildlife, the greatest difficulty has always been the acquisition of samples. Pheasants are difficult to obtain tissue and blood samples from due to their habits and ecological habitats. In general, non-invasive samples of pheasants, such as feathers, feces and eggshells, could be collected for DNA extraction [17,18,19,20], but the quality of DNA extracted with these samples was generally low. It is difficult to develop microsatellite markers from this low-quality DNA using conventional or genome sequencing methods, so most pheasant birds lack available genetic markers. In addition, in the research area of genetic diversity of endangered species, it is difficult to compare the results of different species or different populations of the same species due to the use of different genetic markers. Therefore, the development of universal microsatellite genetic markers is required for most pheasant species in order to facilitate comparative analysis among populations of the same or different species. Meanwhile, it is important that usage of taxon specific universal markers to obtain effective experimental results.

The development of microsatellite loci in pheasants mainly focused on economically valuable pheasant species such as turkey (*Meleagris gallopavo*), indigenous chicken (*Gallus gallus domesticus*), common quail (*Coturnix coturnix*) [21,22,23,24]. Novel microsatellite markers need to be developed for other pheasant birds. Before genome sequencing was widely used, microsatellite markers were mainly developed by using a magnetic bead enrichment method, enriched genomic libraries and cross-species screening of closely related species to screen [25,26,27], which was inefficient and time-consuming. Many microsatellite sequences can be obtained by directly sequencing the genome and rapidly screening from the genome using Next Generation Sequencing (NGS) technology (Illumina MiSeq v3) and bioinformatics software [28,29]. In addition, the genomes of birds are smaller and relatively conserved than those of mammals [30,31], with the potential to develop universal microsatellite loci from the genome.

Based on the genomic data of 8 pheasant species and DNA samples of 14 pheasant species in different clades, this study aimed to develop potential universal microsatellite markers for pheasants using NGS technology and polymerase chain reaction (PCR) to facilitate the comparison of genetic information between species, especially between closely related species. Ultimately, using developed universal microsatellite markers, we evaluated and compared the genetic diversity levels of the Sichuan hill partridge (*Arborophila rufipectus*), blood pheasant (*Ithaginis cruentus*) and buff-throated partridge (*Tetraophasis szechenyii*).

## 2. Materials and Methods

### 2.1. Screening and Localization Analysis of Common Microsatellite Loci Based on the Genome

Three genomes of pheasant species, including the Red Junglefowl (*Gallus gallus*), turkey and Golden pheasant (*Chrysolophus pictus*), were downloaded from the NCBI website (https://www.ncbi.nlm.nih.gov/, accessed on 18 November 2019). Five genomes of the blood pheasant, Chinese monal, Hainan partridge, Sichuan hill partridge and buff-throated partridge were downloaded from our laboratory database [32,33,34,35,36]. The Red Junglefowl annotation file was downloaded from the EMBL database (https://ftp.ensembl.org/pub/release-98/, accessed on 16 November 2019). The whole genome of the Red Junglefowl was used as the template genome, and the remaining 7 pheasant species were used as a comparator genome. Using Krait [37], Microsatellite loci in the genomes of 8 pheasant species were searched and counted. Using CandiSSR [38], common microsatellite loci were selected based on the criteria of single-base repeats ≥12, two-base repeats ≥7, three-base repeats ≥5 and four-base repeats to six-base repeats ≥4 [39,40]. Based on the annotation file, BEDTools (https://www.ncbi.nlm.nih.gov/pmc/articles/PMC2832824/, accessed on 20 November 2019) and g:Profiler (https://www.ncbi.nlm.nih.gov/pmc/articles/PMC4987867/, accessed on 20 November 2019) were used to locate and functionally enrich the common microsatellites [41,42].

### 2.2. Collection of Samples for Screening of Potentially Universal Microsatellites

To further screen for useable microsatellite loci in different pheasant species, we collected 189 samples of 14 pheasant species from various nature reserves, zoos and local research institutions of pheasants in Shaanxi, Tibet, various states and counties of Sichuan for follow-up DNA extraction (Appendix A).

In total, we used genomes of 8 pheasant species to search and count common microsatellite loci, 189 samples of 14 pheasant species to extract DNA, with some crossover of pheasant species in the two steps mentioned above. Finally, a total of 17 pheasant species in 13 genera within Phasianidae were used during the experiments. The 13 genera are almost evenly distributed in the phylogenetic tree of Phasianidae birds (Figure 1).

Animal blood samples were collected by full-time veterinarians from protected areas and other institutions to avoid harming animals to the greatest extent possible, and samples were loaded into vacuum blood collection tubes; muscle and other tissues were taken from unexpectedly dead individuals. All muscle and liver tissue samples in the study were cryopreserved by immersion in anhydrous ethanol, and the blood samples were cryopreserved in centrifuge tubes.

### 2.3. DNA Extraction

DNA was extracted according to the protocol of the animal DNA extraction kit (Trelief Animal Genomic DNA Kit, Chengdu Tsingke Biotech Company, Chengdu, China). All extracted DNA from blood and tissue samples were subjected to 1.5% agarose electrophoresis to assess DNA quality and stored in a −80 °C refrigerator.

### 2.4. Screening of Potential Universal Microsatellite Markers 

Based on the criterion that the microsatellite loci met the number of repeats in the genomes of all 8 species [39,40], the universal microsatellite loci with two-bases, three-bases and four-bases were screened. The flanking sequences at each end were extended by 200 bp, from which primer sequences were generated and sent to Tsingke Biotech Company (Chengdu) to synthesize common primers, and then the PCR conditions were optimized for the synthesized primers (Appendix A for PCR system and Appendix A for PCR procedures are shown in the Appendix A). The PCR amplification products were detected by 1.5% agarose electrophoresis, and images showing bands of the target amplification products without doubt/obscure bands were considered as the best results.

PCR amplification was performed with the optimized primers for each of the pheasants’ DNAs (Appendix A for PCR system and Appendix A for PCR procedures are shown in Appendix A). The PCR amplification products were detected by 1.5% agarose electrophoresis, and potential universal microsatellite markers for pheasants were identified based on the results of gel imaging.

### 2.5. Genetic Diversity Assessment 

Based on the developed potential universal microsatellite markers for pheasants, the corresponding primer sequences were sent to Tsingke Biotech Company (Chengdu) to synthesize fluorescent primers, and the batches were fluorescently labeled with HEX, FAM, or TRAMA at the 5′-end of the forward primers and PCR amplified using fluorescent primers (Appendix A for the PCR system and Appendix A for the PCR procedures are shown in the Appendix A). After the amplification was completed, 4 μL of PCR amplification products were taken from each sample and detected by 1.5% agarose electrophoresis to observe the size of the amplified fragments and the presence of non-specific amplification to assess whether the DNA was successfully amplified.

All PCR products were genotyped on an ABI 3730 DNA Analyzer (Applied Biosystems, Foster City, CA, USA) with GS500LIZ standard and the number of alleles for each sample was determined using GeneMapper v4.0. Genotyping data results were evaluated using the Micro-checker 2.2.3 to check the reliability of the genotyping data [44]. The number of alleles, observed heterozygosity (H_o_), expected heterozygosity (H_e_) and polymorphic information content (PIC) were calculated with Cervus v3.0 [45]. A Hardy–Weinberg equilibrium (HWE) test was conducted with Genepop 3.4 [46]. Finally, the genotyping results of polymorphic loci of microsatellite markers in different samples of the same species were entered into excel sheets to establish microsatellite genotyping databases of the Sichuan hill partridge, Pamuling blood pheasant and Pamuling buff-throated partridge, respectively.

## 3. Results

### 3.1. Common Microsatellite Loci Obtained Using Genome-Wide Data and the Functional Analysis of Localized Genes with Common Microsatellite Loci

Based on genome-wide data from 8 pheasant species, 2328 common microsatellite loci were obtained by adjusting the software screening parameters to exclude Missing Rate (MR) > 50% or single-base microsatellite loci (Appendix A). In this study, only microsatellite loci with a MR of 0 were selected for follow-up study, and a total of 471 common microsatellite loci were finally obtained (Appendix A). 

Common microsatellite loci with varying base pair counts were determined in eight pheasant species (Figure 2A), resulting in a comprehensive tally of 471 common microsatellite loci. Subsequent to the exclusion of recurrent microsatellite loci within distinct genic regions, the positions of the remaining 449 loci were pinpointed (Figure 2B). Among these, 283 loci were situated within genic regions, while 166 loci were identified in intergenic regions.

The gene localization results of common microsatellite loci in the genic region showed that there were 122 genes with common microsatellite loci in the exonic region (Appendix A), of which 115 genes had only one microsatellite locus, six genes (CREBBP, ARID2, FAM155A, KCNB1, NR4A3, PWWP2A) had two common microsatellite loci, and one gene (GRIN2B) had three common microsatellite loci, while the rest of the microsatellite loci are located in the intergenic region. The results of GO functional enrichment of genes with the common microsatellite loci showed that these genes were significantly enriched in these functions that regulate the process of cellular macromolecular biosynthesis (Figure 3), DNA template strand transcription, DNA binding, nucleic acid binding, and other functions that maintain basic cellular biological process, indicating that the common microsatellite loci have some universality among species.

### 3.2. Sample DNA Quality Control

DNA was extracted from 189 blood and tissue samples of pheasants. Based on the results of 1.5% agarose electrophoresis, 81 pheasant samples showed fluorescent bands and 108 samples failed to show fluorescent bands (Appendix A).

### 3.3. Screening Results of Potential Universal Microsatellite Markers

A total of 119 microsatellite loci (Appendix A) from 471 microsatellite loci (Appendix A), including 55 two-base microsatellite loci, 45 three-base microsatellite loci and 19 four-base microsatellite loci, were randomly selected to design the primers. Using the blood pheasant DNA as template DNA, the PCR conditions were optimized by adjusting the annealing temperature and the primers with no product or nonspecific product at all temperatures were thrown away. Then, 81 quality-assessed DNA samples of pheasants were obtained from 14 species distributed among 11 genera within Phasianidae and were used to perform PCR amplification for the common microsatellite primers and detected by 1.5% agarose electrophoresis. Ultimately, 49 pairs of microsatellite primers were obtained that could be successfully amplified in all 14 pheasant species (Appendix A). In total, 22 of the 49 loci were two-bases microsatellite loci, 18 were three-bases microsatellite loci and 9 were four-bases microsatellite loci.

### 3.4. Assessment of Genetic Diversity for Three Pheasants

PCR amplifications were performed on 21 Sichuan hill partridge DNA samples, 19 blood pheasant DNA samples and 14 buff-throated partridge DNA samples using the screened 49 microsatellite loci, and all samples were successfully amplified and genotyped at Tsingke Biotech Company.

Among all the 49 microsatellite loci, 33 loci showed polymorphism in the 21 Sichuan hill partridge DNA samples and 92 alleles were detected in polymorphic microsatellite loci, with the distribution of the allele numbers of each locus ranging from 2 to 5, with an average allele number of 2.79, among which 20 microsatellite loci were within the Hardy–Weinberg equilibrium (*p* > 0.01) (Appendix A). Furthermore, in the 19 blood pheasant DNA samples, 24 loci showed polymorphism and 66 alleles were detected in the polymorphic microsatellite loci, with the distribution of the allele numbers of each locus ranging from 2 to 7, with an average allele number of 2.75, of which 17 microsatellite loci were within the Hardy–Weinberg equilibrium (*p* > 0.01) (Appendix A). Moreover, in 14 buff-throated partridge DNA samples, 22 loci showed polymorphism and 53 alleles were detected at the polymorphic microsatellite loci. The distribution of the allele numbers at each locus was 2~4, with an average allele number of 2.41, of which 17 microsatellite loci were within the Hardy–Weinberg equilibrium (*p* > 0.01) (Appendix A).

Finally, using the accurate typing data, the allele databases of 33 polymorphic microsatellite loci for the Sichuan hill partridge, 24 polymorphic microsatellite loci for the blood pheasants and 22 polymorphic microsatellite loci for the buff-throated partridges were established, respectively (Appendix A).

The count of polymorphic microsatellite loci across all populations of blood pheasant, buff-throated partridge and Sichuan hill partridge was conducted. The results unveiled the presence of 10 universal polymorphic microsatellite loci across all three populations, which were identified as Pha51, Pha53, Pha79, Pha55, Pha80, Pha77, phaA1, PhaA7, PhaG1 and PhaJ8 (see Table 1 and Figure 4A–C).

The mean values of PIC, H_e_ and H_o_ in three species are shown in Figure 4D. The results show that the genetic diversity of the Sichuan hill partridge population in Laojunshan is significantly lower than that of the blood pheasant and buff-throated partridge populations in Pamuling. The difference in genetic diversity between the blood pheasant and buff-throated partridge populations in Pamuling is small, indicating that the universal microsatellite loci could be used among populations of different species.

## 4. Discussion

### 4.1. Universal Microsatellite Screening, Localization and Gene Function

The increased utility of genomic data in research has been accompanied by the widespread use of computational tools based on the development of microsatellite markers. And the development of microsatellite markers involves three main independent steps: microsatellite discovery, primer design and polymorphic survey. To date, a large amount of bioinformatics software has been developed for microsatellite search and development, such as SciRoKo 3.4, msatcommander 1.0.8, MSDB v2.4.3 and GMATA v2.3 [47,48,49,50]. Most of them show excellent performance in genome-wide microsatellite searches, but some of them are limited in the amount of genome sequences they can process. Meanwhile, they lack a uniform output format, which limits downstream analysis and lacks visualization of search results. The Krait used in this study, developed in our laboratory, has a user-friendly graphical interface and fast microsatellite detection speed, supports multiple input and output formats, overcomes the limitations of currently available tools, and can be easily used for searching microsatellite loci. At the step of polymorphism investigation, microsatellite development techniques based on genome sequence data have been shown to be more cost-effective [51,52] and have given rise to software such as MISA v2.1, SSR Primer and SSRPoly 1.0 [53,54,55], but it is difficult for these software to systematically assess microsatellite polymorphisms and to assess the level of confidence for the identification of microsatellites. The CandiSSR used in this study, which provides two confidence metrics including standard deviation and missing rate of the microsatellite repetitions, allows for a systematic assessment of the feasibility of detected microsatellites for subsequent application in genetic characterization. Finally, the experimental results showed that the potential universal microsatellite markers can be used for population genetics of pheasant species.

Microsatellite variation rates are fast and do not strictly follow the stepwise mutation model [56]. Regional genomic factors will influence the variation rate and evolutionary outcome of microsatellites [57]. Therefore, the potential to become a universal microsatellite locus may be related to its location on the genome. According to the location of universal microsatellite loci in this study, microsatellite loci located in genic regions are numerically superior to those located in intergenic regions. There are 283 loci located in genic regions and 166 loci are in intergenic regions (Excluding some microsatellite loci that recurred in different genic regions). In most cases, microsatellite loci located in genic regions, especially in coding regions, have more conserved microsatellite loci due to greater selection pressure [58,59], while microsatellite loci located in intergenic regions are more likely to preserve mutation information and have more mutated microsatellite loci. Furthermore, like any genomic feature that affects phenotype, such variant microsatellites that may have phenotypic consequences will be subject to harsher natural selection [60]. Consequently, the number of universal microsatellites in genic regions are higher than in intergenic regions, which also implies that the longer the evolutionary time span between taxa of organisms is, the greater the proportion of universal microsatellites is in genic regions. In this study, in the large time span of pheasant bird evolution, about 38 million years have passed between the earlier differentiated *Arborophila rufipectus* and the newly differentiated *Chrysolophus pictus* [61], and the number of universal microsatellites in the genic region accounted for about 63% (283/449). Conversely, the proportion of universal microsatellites in intergenic regions would increase in more closely related taxa. However, 166 universal intergenic microsatellite loci were identified in this study, and we suggest they can be more chosen for comparative genetic diversity studies among closely related species.

The genes associated with adaptive evolution in the genome may undergo large mutations during adaptive evolution. The microsatellites located in these genes have also been lost or recreated in some species due to adaptive evolution, and thus these genes contain fewer universal microsatellites, especially among species of distant kinship. Microsatellites contained in some genes related to basic life activities can be conserved in higher numbers and are more likely to be universal microsatellite loci. This was demonstrated by the functional enrichment results of genes containing universal microsatellite loci in this study (Figure 3), for example, the protein encoded by the NR4A3 gene acts as a transcriptional activator; the CREBBP gene is widely expressed and involved in the transcriptional co-activation of multiple transcription factors; the protein encoded by the ARID2 gene plays a role in cell cycle control, transcriptional regulation and chromatin structural modifications.

### 4.2. The Feasibility of Using Universal Microsatellite Markers to Assess the Genetic Diversity of Species

The assessment of genetic characteristics such as population genetic diversity was judged mainly with the help of allele numbers, such as PIC, H_e_ and H_o_ [62,63,64], these parameters can vary considerably depending on the choice of loci. Although there have been some studies on the genetic diversity of rare pheasant species [65,66,67], due to the use of different microsatellite loci, their genetic diversity is difficult to comparatively evaluate. On the contrary, the development of universal microsatellite markers, it is possible to carry out comparative studies on genetic diversity among some closely related species or different populations [68,69], this will help to assess the genetic diversity of species or populations more accurately, to explore the causes of genetic diversity loss and develop conservation responses.

In order to explore the possibility of developing actually usable universal microsatellite markers in pheasants, which have a wide time span of differentiation, we chose 119 out of 449 universal microsatellite loci to perform experimental validation based on 17 species in 13 genera, and these species are scattered in the phylogenetic tree of pheasants, ranging from earlier differentiated clades to newly differentiated evolutionary branches (Figure 1). Ultimately, we developed 49 universal microsatellite loci. Species such as the Sichuan hill partridge, red junglefowl and Reeves’s pheasant are distantly related species [35] and can still be analyzed and compared for genetic diversity using the 49 microsatellite loci developed in our study. Moreover, there is evidence that the applicability of microsatellites among closely related species is higher [69,70,71]. Our results show that 49 universal microsatellite markers can be amplified not only among closely related species, but also among species that have a large phylogenetic range.

Meanwhile, we used 10 out of 49 universal microsatellite loci to comparatively assess the genetic diversity of the Sichuan hill partridge, blood pheasant and buff-throated partridge. The level of genetic diversity of the Sichuan hill partridge in this study was significantly lower than that of the blood pheasant and the buff-throated partridge, which may be due to the Sichuan hill partridge being the first to experience bottleneck effects caused by environmental disasters and overhunting. In the past two decades, the protection work for this species in the Sichuan Laojunshan National Nature Reserve has achieved remarkable results, and the population has been rapidly expanding, but due to the serious fragmentation of the habitat, the isolation of habitat corridors from each other, heterozygous deletions are still evident in this population. Therefore, the increase in the population’s genetic diversity is still slow [67]. Our results reaffirmed that the applicability of potentially universal microsatellite loci developed in this study for comparing genetic diversity among different species.

## 5. Conclusions

Based on genome-wide data among 8 Phasianidae species, 471 common microsatellite loci were screened. Among these loci, 119 microsatellite loci were selected for primer design and optimization. The optimized microsatellite primers were applied to PCR amplifications of the genomic DNA of 14 pheasant species, and 49 universal microsatellite loci were developed. These universal microsatellite loci were successfully applied to genetic diversity assessments and comparisons for three pheasant birds, namely the Sichuan hill partridge, blood pheasant and buff-throated partridge. Since the pheasant species used for this study are large-scale and scattered evenly in different evolutionary branches of the phylogenetic tree, these universal microsatellite loci can be used not only as a standardized evaluation criterion for population genetics of pheasants, but also for the comparison of genetic data among different species of pheasants to establish a more comprehensive genetic profile of pheasants. However, given the limited number of pheasant species for which the applicability of universal microsatellite markers has been verified, the results of our experiments should be viewed with caution and could continue to optimize universal microsatellite markers.

## Figures and Tables

**Figure 1 animals-13-03601-f001:**
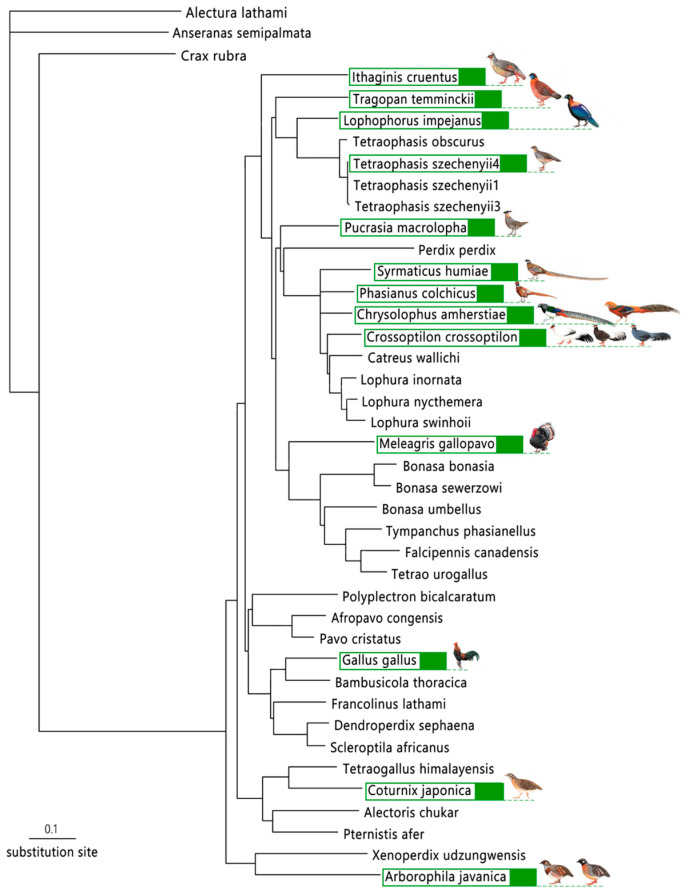
Phylogenetic tree of the 17 pheasant species. The bird images represent the 17 pheasant species, distributed among 13 genera, used in this study. Phylogenetic tree base map quoted from the complete mitochondrial genomes of galliform birds [43].

**Figure 2 animals-13-03601-f002:**
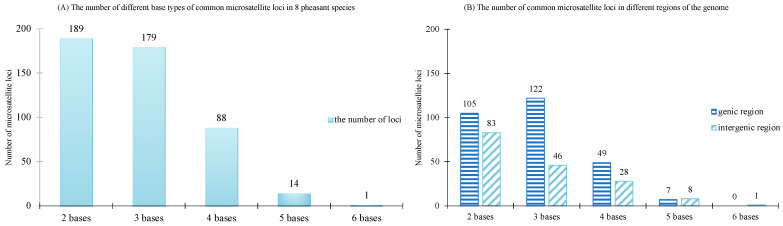
Number of common microsatellite loci of different base types (**A**) and in different regions of the genome (**B**). There are 2 bases, 3 bases, 4 bases, 5 bases and 6 bases on behalf of two-base repeats microsatellite loci, three-base repeats microsatellite loci, four-base repeats microsatellite loci and six-base repeats microsatellite loci.

**Figure 3 animals-13-03601-f003:**
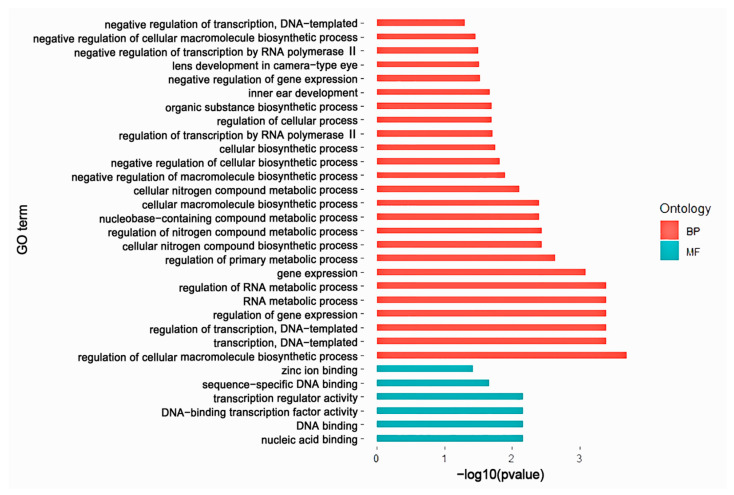
The GO enrichment maps of the universal microsatellite loci. Biological process (BP) represents the pathways and biochemical processes comprising multiple gene products. Molecular function (MF) represents the molecular function of the product of a gene.

**Figure 4 animals-13-03601-f004:**
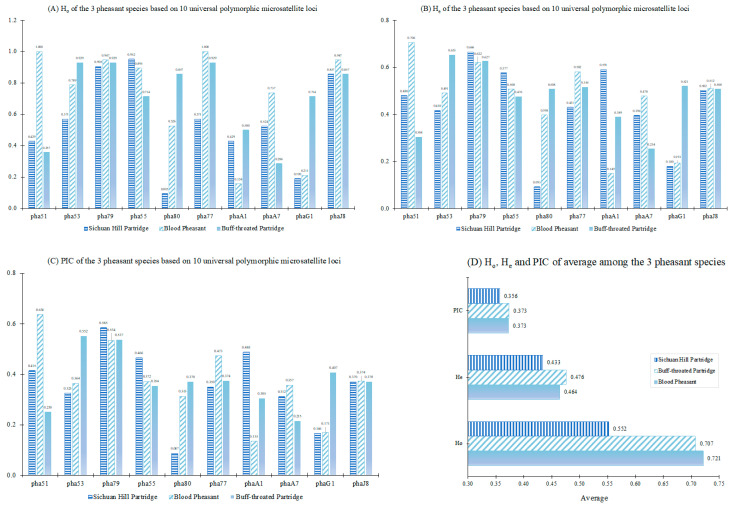
Comparison of H_o_ (**A**), H_e_ (**B**), PIC (**C**) and mean values of H_o_, H_e_ and PIC (**D**) among the populations of Blood pheasant, buff-throated partridge and Sichuan hill partridge based on 10 universal microsatellite loci. Pha51, Pha53, Pha79, Pha55, Pha80, Pha77, phaA1, PhaA7, PhaG1, PhaJ8 represent 10 different universal polymorphic microsatellite loci.

**Table 1 animals-13-03601-t001:** H_o_, H_e_ and PIC analysis based on universal polymorphism microsatellite loci in 3 populations.

Loci	H_o_			H_e_			PIC		
	Sichuan Hill Partridge	Blood Pheasant	Buff-Throated Partridge	Sichuan Hill Partridge	Blood Pheasant	Buff-Throated Partridge	Sichuan Hill Partridge	Blood Pheasant	Buff-Throated Partridge
pha51	0.429	1.000	0.357	0.480	0.706	0.304	0.415	0.636	0.250
pha53	0.571	0.789	0.929	0.418	0.491	0.653	0.325	0.364	0.552
pha79	0.905	0.947	0.929	0.666	0.622	0.627	0.585	0.534	0.537
pha55	0.952	0.895	0.714	0.577	0.508	0.476	0.466	0.372	0.354
pha80	0.095	0.526	0.857	0.093	0.398	0.508	0.087	0.313	0.370
pha77	0.571	1.000	0.929	0.431	0.582	0.516	0.350	0.473	0.374
phaA1	0.429	0.158	0.500	0.591	0.149	0.389	0.488	0.135	0.305
phaA7	0.524	0.737	0.286	0.396	0.478	0.254	0.312	0.357	0.215
phaG1	0.190	0.211	0.714	0.180	0.193	0.521	0.166	0.171	0.407
phaJ8	0.857	0.947	0.857	0.502	0.512	0.508	0.370	0.374	0.370
Average	0.5523	0.7210	0.7072	0.4334	0.4639	0.4756	0.3564	0.3729	0.3734

Note: Sichuan hill partridge, blood pheasant and buff-throated partridge are the 3 pheasant species used in this study.

## Data Availability

The data presented in this study are available in the article or Appendix A.

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
