# Peer review of "Development and Application of Potentially Universal Microsatellite Markers for Pheasant Species"

_animals, 2023, doi:10.3390/ani13233601_

Round 1

Reviewer 1 Report

Comments and Suggestions for Authors

Dear authors 

Although I found the article very interesting, I would like to suggest a few improvements. 

In topic 1. Introduction page 1, line 42,43, 44 and 45

Use a phylogeny based on genetic data and not just morphological, the authors cite in line 122 and 123 a phylogeny based on mitochondrial DNA. Why not use a nuclear DNA phylogeny?

In the topic Materials and Methods

The sample number is not clear. It's confusing: Genome of 8 species (common microsatellite locci). Samples from 14 species? Genomic data from 17 species? Rewrite

In topics 3. Results

On line 181 where it says:

The number of common microsatellite loci with a different type of base numbers in 8 pheasant species is shown in Figure 2(A), with a total of 471 common microsatellite loci. After excluding some microsatellite loci that recurred in different gene regions, the positions of the remaining 449 loci are shown in Figure 2(B). Among them, 283 loci are in gene regions and 166 loci are in intergenic regions.

Replace with:

Common microsatellite loci with varying base pair counts was determined in eight pheasant species (Figure 2-A), resulting in a comprehensive tally of 471 common microsatellite loci. Subsequent to the exclusion of recurrent microsatellite loci within distinct gene regions, the positions of the remaining 449 loci were pinpointed (Figure 2-B). Among these, 283 loci were situated within gene regions, while 166 loci were identified in intergenic regions.

In line 247 where it says:

The microsatellite loci that showed polymorphism in all populations of Blood Pheas ant, Buff-throated Partridge and Sichuan Hill Partridge were counted. The results showed that there were 10 universal microsatellite loci with polymorphism in all 3 populations,  namely Pha51, Pha53, Pha79, Pha55, Pha80, Pha77, phaA1, PhaA7, PhaG1, PhaJ8. Comparative analysis among 3 species using the 10 universal polymorphic loci is shown in Table 1. The results of the apparent difference in the values of the genetic diversity parameters are shown in Figure 4(A), Figure 4(B) and Figure 4(C)

Replace with:

The count of polymorphic microsatellite loci across all populations of Blood Pheasant, Buff-throated Partridge, and Sichuan Hill Partridge was conducted. The results unveiled the presence of 10 universal polymorphic microsatellite loci across all three populations, which were identified as Pha51, Pha53, Pha79, Pha55, Pha80, Pha77, phaA1, PhaA7, PhaG1, and PhaJ8 (see Table 1 and Figure 4(A), Figure 4(B), and Figure 4(C)).

In line 254 add in the table title: (Ho), expected heterozygosity (He) and polymorphic information content (PIC)

In line 265 add in the figure caption: (Ho), expected heterozygosity (He) and polymorphic information content (PIC)

I still suggest that an English specialist reviews and corrects the writing.

Author Response

Dear Reviewer,

Thanks for providing us this great opportunity to resubmit a revised version of our manuscript. We appreciate the detailed and constructive comments provided by the reviewer. Your suggestions have enabled us to improve our work. Now, we have carefully revised the manuscript by incorporating all the suggestions by the reviewer. In this revised version, changes to our manuscript within the document were all highlighted by using the “Track Changes” function. And the following is an explanation of the revisions we have made.

In topic 1. Introduction

  1. Why not use a nuclear DNA phylogeny?

(1) Both nuclear gene sequences and mitochondrial genomes can be used to construct phylogenetic trees, and the phylogenetic trees constructed are very close to the time of differentiation.

(2) Adequate mitochondrial genome studies were carried out in this laboratory. The genomic data of Blood Pheasant, Chinese Monal, Hainan Partridge, Sichuan Hill Partridge, and Buff-throated Partridge used in the experiments were obtained from our laboratory (References 32-36).

In topic 2. Materials and Methods

  1. On the question of ambiguity in the presentation of the sample

We've done a rewrite.

In topics 3. Results

  1. 247 Contents of line 181 and line 247 need to be replaced.

We've finished replacing.

  1. In line 254 add in the table title: (Ho), expected heterozygosity (He) and polymorphic information content (PIC); In line 265 add in the figure caption: (Ho), expected heterozygosity (He) and polymorphic information content (PIC).

We're done with the changes.

In addition, we have corrected some of the writing.

It is the content of all our responses. We deeply appreciate your suggestions of our manuscript. If you have any other queries, please don’t hesitate to contact me at the address below. Thank you and best regards.

Yours sincerely,

Daxin Xie

Author: Daxin Xie; E-mail: [email protected]

Reviewer 2 Report

Comments and Suggestions for Authors

The authors present an interesting study on development and application of microsatellites markers. Study design and research questions are clearly described. In this sense, it is easy to understand the aim of this study. The bright side of the manuscript is that to provide practical details on related topic and new microsatellites which can be used by many people in related fields. In this context, the study contributes to different fields. Only minor concerns are raised. Therefore, I would like to make some suggestions to improve the quality of the paper as below:

General comments

The Discussion section should be enriched with a more theoretical interpretation and relate the present results with additional concepts. For instance, the study results can be discussed in the framework of development of microsatellite markers comparing the results of the study with similar studies in the broader context.

Some parts of the manuscript are not easy to understand (mentioned below in specific comments). There are some long sentences and wordiness. This situation disrupts the flow of the subject and the continuity of the reading. Because of this reason, authors should reconsider writing some parts of the manuscript.

Abstract

Line 26: assessment of genetic diversity -> assessment of genetic diversity and planning effective conservation actions

Introduction

Lines 41-45: Please just add 2-23 sentences that explain the evolution of Galliformes and/or Phasianidae. After that, please explain the the conservation status of members of Phasianidae on global level.

Line 53-55: “Due to illegal hunting, coupled with habitat decline and loss, many pheasant species are on the verge of extinction, which shows the seriousness of the conservation situation of pheasant species and the need to promote conservation research.” Please add references.

Line 57-60: “Microsatellite markers have been widely used in the scientific areas of molecular ecology and population genetics of endangered animals due to their wide distribution in genome, rich polymorphic information content, and co-dominant inheritance ” Please add this study here (DOI: 10.1080/14772000.2019.1691673 and 10.3390/biology12030401).

Lines 67-70: Authors may also say that usage of taxon specific universal markers is important to for effective results.

Lines 92-93: “The results showed that the potential universal microsatellite markers can be used for population genetics of pheasant species.” Please move this sentence to discussion or conclusion section.

Materials and Methods

Lines 99-101: “Five genomes of the Blood Pheasant, Chinese Monal, Hainan Partridge, Sichuan Hill Partridge, and Buff-throated Partridge were downloaded from our laboratory database.” Please add references if any published papers are available related to these genomes.

Discussion

Please add a single paragraph or more about the microsatellite development process to discussion section. You may compare your effort with similar studies by comparing the methods, time, accuracy, and sampling.

Line 284: high taxonomy? Do the authors mention “higher taxonomic level”? Please rephrase or explain.

Lines 295-296: “Genes in the genome that have been associated with adaptive evolution may have large mutations during adaptive evolution across a large time span.” Please rephrase this sentence.

Lines 311-315: “Although there have been some studies on the genetic diversity of some rare pheasant species, for instance, for the Buff-throated Partridge in Pamuling population [43,44], and for the Sichuan Hill Partridges in Laojunshan [45]. However, due to the use of different microsatellite loci, their genetic diversity is difficult to comparatively evaluate.” Please merge these 2 sentences. On the other hand, there are many microsatellite markers which can be used for the comparative analysis. However, these markers are not taxon specific so you may also emphasise the need for taxon specific markers.

Line 316: Authors may say “universal microsatellites and/or taxon specific microsatellites instead of “some universal microsatellite”.

Conclusion

Lines 346-356: The limitations of the study should be given in the conclusion section. Moreover, authors should explain the main contribution of their results (producing the universal or taxon specific microsatellite markers) to the field with 1-2 sentences. 

Comments on the Quality of English Language

Some parts of the manuscript are not easy to understand. There are some long sentences and wordiness. This situation disrupts the flow of the subject and the continuity of the reading. Because of this reason, authors should reconsider writing some parts of the manuscript.

Author Response

Dear Reviewer,

Thanks for providing us this great opportunity to resubmit a revised version of our manuscript. We appreciate the detailed and constructive comments provided by the reviewer. Your suggestions have enabled us to improve our work. Now, we have carefully revised the manuscript by incorporating all the suggestions by the reviewer. In this revised version, changes to our manuscript within the document were all highlighted by using the “Track Changes” function. And the following is an explanation of the revisions we have made.

In topic 1. General comments

We have enriched the discussion with a section on microsatellite screening software. Because the article focuses on the development and use of microsatellite markers, we compare the advantages and disadvantages of commonly used screening software in existing studies. Besides, we rewrote some of the ambiguities in the article.

In topic 2. Abstract

Line 26: We've changed "assessment of genetic diversity" to "assessment of genetic diversity and planning effective conservation actions".

In topic 3. Introduction

Line 41-45: We have explained the evolution of the Galliformes. Then, the conservation status of birds of the pheasant family is described on a global scale.

Line 53-55: “Due to illegal hunting, coupled with habitat decline and loss, many pheasant species are on the verge of extinction, which shows the seriousness of the conservation situation of pheasant species and the need to promote conservation research.” We have added relevant references.

Line 57-60: “Microsatellite markers have been widely used in the scientific areas of molecular ecology and population genetics of endangered animals due to their wide distribution in genome, rich polymorphic information content, and co-dominant inheritance.” We have added this study. (DOI: 10.1080/14772000.2019.1691673 and 10.3390/biology12030401).

Lines 67-70: We have optimized the expression.

Lines 92-93: “The results showed that the potential universal microsatellite markers can be used for population genetics of pheasant species.” This sentence has been added to the discussion section.

In topics 4. Materials and Methods

Lines 99-101: “Five genomes of the Blood Pheasant, Chinese Monal, Hainan Partridge, Sichuan Hill Partridge, and Buff-throated Partridge were downloaded from our laboratory database.” We have added references from our laboratory related to these genomes.

In topics 4. Discussion

We have added a single paragraph about the microsatellite development process to discussion section. The main comparison is between the advantages and disadvantages of various microsatellite screening software.

Line 284: We have removed this part of the misrepresentation.

Lines 295-296: “Genes in the genome that have been associated with adaptive evolution may have large mutations during adaptive evolution across a large time span.” We've rewritten the sentence.

Lines 311-315: We merged these two sentences and emphasized the necessity to develop taxon-specific markers.

Line 316: We've changed " some universal microsatellite markers " to " universal microsatellite markers".

In topics 5. Conclusion

Lines 346-356: We have described the limitations of the study and explained the main contributions of the research results in conclusions.

In addition, we have corrected some of the writing.

It is the content of all our responses. We deeply appreciate your suggestions of our manuscript. If you have any other queries, please don’t hesitate to contact me at the address below. Thank you and best regards.

Yours sincerely,

Daxin Xie

Author: Daxin Xie; E-mail: [email protected]

Round 2

Reviewer 1 Report

Comments and Suggestions for Authors

The manuscript was sufficiently improved to justify publication in Animals.